# Natural Products against Sand Fly Vectors of Leishmaniosis: A Systematic Review

**DOI:** 10.3390/vetsci8080150

**Published:** 2021-07-30

**Authors:** Michela Pugliese, Gabriella Gaglio, Annamaria Passantino, Emanuele Brianti, Ettore Napoli

**Affiliations:** Department of Veterinary Sciences, University of Messina, 98168 Messina, ME, Italy; mpugliese@unime.it (M.P.); ggaglio@unime.it (G.G.); ebrianti@unime.it (E.B.); enapoli@unime.it (E.N.)

**Keywords:** sand fly, vectors, vector borne diseases, repellence, natural products, plant-based products

## Abstract

Leishmaniosis is a vector-borne disease transmitted to animals and humans by the bite of blood-sucking phlebotomine sand flies. These small insects play a crucial role in the diffusion of the disease. To date, the sole strategy recognized for the prevention of leishmaniosis is the use of topical repellent compounds against sand fly bites. Several synthetic insecticides and repellents have been developed; however, the wide and unprejudiced use of these formulations have led to the loss of their effectiveness and the development of resistance phenomena. Moreover, some of these synthetic repellents have severe detrimental effects on the environment and could represent a serious threat to both animal and human health. Recently, an increased interest in the research on alternative approaches to sand fly control has been expressed. In this study, we systematically reviewed the efforts of the scientific community to individuate a phytochemical alternative for the control of sand fly species recognized as vectors of *Leishmania* spp. Based on literature research using different electronic databases, a total of 527 potentially relevant studies were screened and narrowed down to a final 14 eligible scientific reports. Our analysis suggests that although there is a rapidly growing body of literature dedicated to botanical insecticides and repellents against sand fly vectors of *Leishmania* spp., much of this literature is limited to in vitro studies conducted in laboratory conditions, and only a few of them investigated the repellency of plant-based products. These studies highlighted that natural compounds display a really short period of action and this significantly limits the use of these products as an alternative to chemical-based repellents.

## 1. Introduction

Leishmaniosis is a zoonotic disease regarded as one of the most common vector-borne diseases (VBDs) throughout the world [1]. To understand the relevance of this VBD, it is enough to underline that the WHO (2010) [1] estimated about 50,000 to 90,000 new cases of leishmaniosis annually, which occurs in humans worldwide and remains one of the top parasitic diseases with a potential for outbreak and mortality. Leishmaniosis is transmitted to animals and humans through the bite of blood-sucking phlebotomine sand flies (Diptera: Psychodidae) [2] belonging to the genera *Phlebotomus* and *Lutzomyia* in the Old and New World, respectively [3,4]. These insects play a crucial role in the epidemiology of the disease, being the only proven vectors of *L. infantum;* their bites are the only way to transmit the disease [2]. Phlebotomine sand flies are small insects (1.5–3.5 mm in length) and only the female needs to feed on blood in order to gain all the nutrients necessary for egg production [2]; these blood meals are taken on many vertebrate hosts (i.e., reptiles, birds, and mammals) since they are considered opportunistic and general feeders [5]. The domestic dog, *Canis lupus familiaris,* has been regarded as the reservoir of the infection for a long time [6]; however, other domestic and wild mammals have recently been recognized as co-protagonists in the maintenance of the disease [5]. Over the last few decades, many efforts to limit the spread of leishmaniosis have been made and several strategies have been formulated to improve its control and the prevention; some of these efforts were focused on the treatment or the control of the causative agents (i.e., vaccination), but the majority were focused on the control of the vectors, which currently represents the sole strategy for the prevention of the disease [7]. In the past, control measures aimed at reducing vector populations in the environment by means of insecticides have been tried; however, the environmental treatment against the adult stages has a transitory effect and is unsustainable in the long term for several technical and economic reasons [7]. While the control of immature stages of sand flies is considered unpractical due to the wide variety of microhabitats (e.g., tree roots and holes, animal burrows, leaf litter, manure, holes, and crevices in walls) favorable for the breeding of larvae and pupae [4,7], the economic cost of this kind of approach is also inconvenient when compared to its real benefit.

So far, the sole strategy recognized for the prevention of leishmaniosis is the use of topical repellent compounds in different formulations for individual protection against the sand fly bites [7,8]. The concept of repellence is well-defined in the etymology of its name—the term “repellent” derives from the ancient Latin “*repellere*”, which means “to reject”. A repellent can be defined as a volatile substance, natural or synthetic, which induces an insect to move towards an opposite direction, nullifying the attractive stimulus represented by an animal or a human [9,10,11]. The concept of repellence was developed by the first humans many years ago; in fact, since the dawn of history, humans have had to drive insects away from habitation and for this purpose, they have had to develop a different strategy. The use of smoke, typically obtained from the burning of plants, was undoubtedly the first and the most common method used for repelling insects [12]; equally pioneering was the use of plant and plant-based products on animals and humans for insect repellence when applied onto the hair, skin, or clothes [12].

In many parts of the world, plant-derived products (i.e., essential oils or plant extracts) were traditionally used to repel and kill insects [13]. Plant-based insecticides tend to have a broad spectrum of activity and are normally safe for both animals and the environment [14]. Indeed, crude plant-extract and essential oils were widely used as insecticides up to the beginning of WWII in the 1940s when organic laboratory-synthesized insecticides became available [15,16]; in particular, dichloro-diphenyl-trichloro-ethane (DDT) was introduced as a repellent and insecticide in 1939 [12]. However, this compound has negative effects on the environment and human health, and has been banned in most countries of the world [17]. Nevertheless, DDT is still routinely used in some developing countries, most of them in Africa, to fight mosquitoes that carry malaria [18]; in India, this compound is used to control the adult stage of sand flies [19] despite demonstrating a high resistance to DDT [20].

Pyrethroids have been developed in Europe since the 1930s [21], showing minimal toxicity for mammalian; these products are the synthetic analog of pyrethrum. Pyrethrum is a natural oil derived from the flowers of the plant *Chrysanthemum cinerariifolium* [15]. Among the synthetic analogs of pyrethrum, the permethrin, first synthesized in the United States in 1972 [22] and initially registered for agricultural use in 1979, became a cornerstone for the prevention and control of important VBDs in a few years’ time. The use of this molecule in different formulations is effective in controlling the vectors of malaria [2], as well as useful against the sand fly vectors of *Leishmania* [3,23,24]. From these precursors, several synthetic insecticides and repellents have been developed in the last decades; used alone or in combination, they have contributed to the protection of humans and animals against insects and the related VBDs. However, the wide and unprejudiced use of these formulations have led to the loss of their effectiveness and the development of resistance phenomena [25,26]. Moreover, considering the restrictions on the use of some insecticides and repellents because of their effects on both animal and human health [27] as well as on the environment [28], the search for natural repellents as an alternative to chemical-based products has been pushed forward in the last years. This colossal turnaround, intended as a return to the use of plant-based products against arthropods, has brought about extensive research into ticks [25,26,29] and mosquitoes [13,30,31,32,33,34,35,36,37], while the use of plant-based products against sand flies has been less investigated.

In this study, we systematically reviewed the efforts of the scientific community to individuate a phytochemical alternative for the control of sand fly species recognized as vectors of *Leishmania* spp.

## 2. Material and Methods

### 2.1. Data Sources and Search Strategy

To meet the objectives of this article, all the eligible studies on the repellency effects of plant-based products against sand flies published from January 1990 to June 2020 were systematically searched using electronic databases, which include PubMed, Medline, Google Scholar, and BMJ (British Medical Journal), using the following Medical Subjects and keywords:

Plant [Title/Abstract], plants [Title/Abstract], extract [Title/Abstract], extracts [Title/Abstract], essential oil [Title/Abstract], essential oils [Title/Abstract], Insect repellent [Title/Abstract], repellent [Title/Abstract], repellence [Title/Abstract], repellency [Title/Abstract] and sand flies or sand fly [Title/Abstract]; moreover, a manual search was conducted using references from retrieved studies.

### 2.2. Inclusion and Exclusion Criteria

For the present systematic review, we only considered and included publication that met the following inclusion criteria:The publication was in English (i.e., at least the abstract);The full text was available;Inspected the effects of plant-based products against sand fly vectors of *Leishmania* spp.;Reported the percentage of repellency and/or complete protection time and/or insecticide efficacy;Were original studies conducted in the laboratory and/or field conditions.

Moreover, we excluded the following from the present review: articles without an available full text, books, documents, republished data, conference papers, reviews, systematic reviews, and meta-analyses; articles that were conducted on non-target insect species. 

All the articles identified were screened for title and abstract independently by two different authors (EN, MP); for those articles considered suitable, based on title and abstract and that met the inclusion criteria and did not fall within the exclusion criteria, the full text were further evaluated independently by the two authors to be included in systematic review.

## 3. Results

The available literature was searched, as reported above, and a total of 511 potentially relevant studies were identified in electronic databases; 16 were obtained from a manual search according to the references of the retrieved studies and in grey literature. Figure 1 presents the flowchart of the preliminary assessment from which we extrapolated the number of scientific reports on plant-based products tested against sand flies. In particular, the majority of scientific reports were identified on Pubmed and Google Scholar (i.e., 295 and 200, respectively), followed by Medline and BMJ (i.e., 11 and 5, respectively).

Among the 527 potentially relevant studies, we identified 4 replicates; 456 that fell under the exclusion criteria or did not fulfill the inclusion criteria, 409 of which were not focused on the target species; 32 were reviews; 11 articles were not in English; and 4 did not have a full text available. Therefore, a total of 67 full texts were evaluated for inclusion in the review. Figure 1 illustrates the flow of the study selection process. Of the 67 articles evaluated, 53 were excluded after the evaluation of the full text because 10 were similar to a review article, 41 tested their efficacy on non-Leishmania vectors, 3 tested a compound that did not fall under the inclusion criteria (i.e., 1 pheromone, 1 nanostructure hydrogels, and 1 KBR 3023 or icardin); a total of 14 articles included in the present manuscript were about plant-based products against sand flies.

Table 1 summarizes the scientific reports that fulfilled the inclusion criteria. In particular, seven were conducted in vitro [38,39,40,41,42,43], while only five were conducted in vivo [44,45,46,47,48] in laboratory conditions, and two explored both conditions [49,50,51]. As reported in Figure 2A, the number of scientific reports was unevenly distributed in the study period (i.e., from 1990 to 2020), although a positive trend was observed. In particular, the majority of the publications (11/14; 78.6%) were produced in the last decade of the time window analyzed.

Figure 2B presents the geographical origin of the studies reported herein. Out of six continents, 42.86% (i.e., 6/14) of the studies were produced in South America, with three in Venezuela [42,43,49] and three in Brazil [39,40,48]; of the 28.57% (i.e., 4/14), three were produced in India [38,44,45] and one in Iran [47]; 21.43% of the studies were produced in Africa (i.e., two in Kenya [41,51] and one in Ethiopia [50]); only one was produced in Europe (i.e., in Italy) [46].

Going into the specifics of the plant-based products used in the scientific studies, a big variety of the plants tested belongs to twenty-five different species and eleven different families.

Table 2 summarizes the plant and the relative family. In particular, the majority of the species belong to the *Asteraceae* (i.e., 9/25; 36%) and the *Myrtaceae* (i.e., 5/25; 20%) families. However, the majority (i.e., 28.57%) of the studies were focused on the *Meliacee* family, and in particular, on *A. indica* (i.e., the neem oil). The botanical product is related not to the geographic origin where the study was performed but to the sand fly species. In fact, the plants that were mainly tested from Africa and Asia were *A. indica*, against *Ph. orientalis* [50] and *Ph. papatasi* [38,44,45,47], respectively. While in South America, the largest number of plants was tested, all against the sand fly species belonging to the genus Lutzomyia. The efficacy of the different formulations of these plant-based products was tested; in particular, eight studies were used in essential oils [38,40,42,43,46,47,48,49,51], three in hexane-extracted oil [44,45,50], and four in ethyl acetate, methanol, or aqueous extract [39,41,42,43].

The majority of the studies herein reported investigate the insecticide effects of plant-based products (i.e., 8/14; 57.28%); however, especially in the last decades of the time-window considered, the attention of researchers was focused mainly on the repellent activity of plant-based products against sand flies (i.e., 6/11, 54.54%) [39,45,46,47,50,51].

The efficacy of the tested compounds, which needs at least 24 h to reach the maximum in all the selected studies, normally persists only for a limited period, although only a few studies consider this aspect.

All the plant-based products investigated as repellents showed high efficacy (i.e., >80%), although these products demonstrated a short period of action. For instance, Rojas and Scorza (1991) [49] reported a repellency timespan of 30 min (i.e., 70% of efficacy) for *C. medica* against *Lu. youngi*, while the repellency of *P. caeruleocanum and C. zylancnicum* against *Lu. mingonei* [39] was longer (i.e., 3 h). The neem oil, at concentrations of 2% and 5%, showed a repellency of 99.57% against *Ph. Orientalis*, and 99.2% (i.e., 5%) and 92% (i.e., 2%) against *Ph. bergoroti* [50] for 8 h and 24 min, respectively. Myrtle essential oils of *T. minuta* and *C. citratus* demonstrated repellent activity against *Ph. dubosqui* of 88.89% and 100%, respectively, for 3 h. 

The effect of plant-based products, as an insecticide or as a repellent, was assessed on seven different sand fly species; in particular, three of the species belong to the genus *Lutzomyia* (i.e., *Lu. longipalpis*, *Lu. mingonei*, and *Lu. youngi*, all regarded as vectors of *L. infantum*) and four to the genus *Phlebotomus* (i.e., *Ph. papatasi*, the main vector of *L. infantum*; *Ph. dubosqui* and *Ph. bergoti*, vectors of *L. major*; *and Ph. orientalis*, vector of *L. donovani*).

## 4. Discussion

This study systematically reviewed the available literature published between 1990 and 2020 on the effect—insecticide or repellent—of plant-based products against the sand fly species recognized as vectors of *Leishmania* spp., thus providing evidence that only a few attempts were made to use natural products against these vectors.

Starting from their development before WWII in the 1940s [15,16], the use of synthetic insecticides and repellents has significantly increased in both animals and humans in order to prevent losses directly or indirectly related to blood-sucking arthropods and the related VBDs (i.e., lives, economic resources and money used to cure the related diseases) [26]. The use of a large number of synthetic compounds has inevitably led to the creation of arthropod species that are resistant to these chemicals [26]; moreover, it has been stated that this wide use of chemical products represents a serious threat to both human and animal health (i.e., through residuals in foods and feeds) as well as a risk for the environment and the preservation of biodiversity [15,52,53].

Therefore, the use of the so-called “green chemistry”, based on the use of microbial and plant products, selected metabolites as well as green synthesized nanostructures, have been widely explored [54]. Plant-based insecticides or repellents proved to have a wide-spectrum of activity against hematophagous arthropods and to be highly safe [15]. Several plants (i.e., more than 2000) have a potential insecticidal effect; however, the properties of just a few of these were explored. Up-to-date plant-based insecticides represent only 1% of the world’s pesticide market.

A recent study, corroborated by the findings herein reported, demonstrated that the majority of these “green chemistry” products were tested on mosquitoes (i.e., 668) [30,54], but only a little attention was given to other blood-sucking arthropods. In particular, after consolidating the publications on the toxic and repellent activity of plant-based products against biting midges, black flies, horse and deer flies, stable flies, tsetse flies, lice, bed and kissing bugs, fleas and sand flies—only 106 scientific reports were identified, and only six of which investigated the activity of these products against sand flies [54]. In the present study, considering the so-called “*grey literature*” revealed more publications on the topic, and a total of 67 scientific reports were identified although only 14 fully satisfied the inclusion criteria proposed. However, it should be noted that in general, the research focused on the development of botanical products against sand flies just started in the recent years [54,55], and as herein reported, an increasing trend in the number of publications has been observed. As reported elsewhere, the growth in the field of botanical insecticide research has been explosive, from only 61 papers in 1980 to 1207 in 2012 [55]. In the same manner, 78.6% of the studies analyzed systematically in the present study were conducted between 2001–2017. This increased interest in botanical products corresponds to an increasing need for an alternative to chemical products. However, many factors that could, in some way, limit the study of natural products against the adult stages of sand flies persist. The major limitation is probably the difficulties of maintenance and availability of sand fly colonies [47]; this last finding could explain the limited number of scientific reports that investigate the toxicity and the repellence of botanical product against adult stages of sand flies. 

It is thus not surprising that the majority of the analyzed studies herein reported were conducted in countries with a high incidence of visceral and cutaneous human leishmaniosis such as Brazil, India, and Ethiopia [1,56], where the use of botanical products is inherent in the cultural tradition [13,39].

India, China, and Brazil have been defined as the ‘big 3’ in the research of botanical insecticides [55]. In fact, these three nations produced the greatest number of publications on plant-based products (i.e., the 40.9% of botanical insecticide articles published in 2012). However, in the present study, no scientific publication produced in China has been identified. This last finding could be related to the fact that leishmaniosis is a sanitary problem controlled in most provinces in China, and it remains fundamentally uncontrolled only in some northwestern provinces and autonomous regions (i.e., Sichuan Province, Gansu Province, and the Xinjiang Uygur Autonomous Region) [57]. With regard to the research on botanical products against sand flies, it is important to mention, along with the ‘big 3’, the research activity produced in some African (Ethiopia and Kenya) [41,50,51] and Middle Eastern countries (Iran) [47]; on the other hand, the number of articles from developed countries is negligible, with only one report found in Europe (i.e., Italy) [46]. The absence of scientific reports produced in Europe may be due to a lesser extent to the scarce use of products of natural origin, and primarily to the fact that in southern Europe, leishmaniosis is mainly a health problem in dogs. The use of repellents or insecticides of natural origin on dogs is limited by many factors such as the smell and the period of effectiveness (i.e., from 30 min to a few hours) of these products, as well as the compliance of the owner.

It is worth noting that the choice of botanical product is related to the availability of the plant in the surroundings; in fact, in different studies, the herbs were taken directly from the fields and the essential oils/or extracts were produced in the laboratory. On the other hand, it is evident that the choice of the product was mainly influenced by the sand fly species. Neem oil, or *A. indica,* was the main plant tested in both Africa and Asia against the sand fly that belongs to the *Phlebotomus* genus (i.e., *Ph. orientalis, Ph. papatasi* and *Ph. bergeroti*) [38,44,45,50]. The *A. indica* essential oil at 2% or 5% has been widely used as an ethnic low-cost alternative protection against sand fly bites, particularly in regions where insecticides are not applied for administrative and economic motivations. It has been shown that the oil at both 2% and at 5% is protective against bites of female sand flies. Its efficacy was proven to be higher for *Ph. argentipes* than against *Ph. papatasi*, representing the main vector of visceral and cutaneous leishmaniasis in India [44,45]. Similar efficacy was demonstrated for *Ph. orientalis* and *Ph. bergeroti* in Ethiopia [50].

Particularly noteworthy is the fact that only one scientific report investigates the repellent effect of lemongrass against *Ph. dubosqui* [51]. The extract of this last plant has been widely used as a repellent against mosquitoes in different formulations (i.e., spray, oils, and candles) [58,59,60].

In contrast to the sand flies belonging to the genus *Lutzomyia*, there are different plant-based products that have been tested, most of which were based on essential oils or extracts of plants that belong to the Asteraceae family [39,42,43], and to a lesser extent to the *Myrtaceae* family [39,40]. However, the only products that in some way showed promising results were the *A. ovata* aqueous extract [39] against *Lu. Longipalpis,* and the *M. greenmaniana* essential oil against *Lu. migonei* [42]. The other botanical products showed a low or no efficacy.

Independently of the botanical products investigated, the majority of the studies were focused on essential oils, that is, as already observed for other vectors as mosquitoes [30,31,32,33,34,35,36,37] and ticks [25,26,29]. Essential oils were the formulation that provided the more interesting results. This formulation, in the majority of the studies herein reported, reaches an efficacy of up to 100% [40,41,42,43], while the amount of protection time it offers against sand fly bites was longer when compared to other formulations (i.e., crude plants extract).

Unfortunately, as stated elsewhere for mosquitoes [55], only a few studies have simultaneously compared the effects of botanical and synthetic products [47]. Therefore, making a real comparison between botanical and synthetic products is not easy as the insecticide and repellent effects of a given product could be related to multiple factors related to the compound (i.e., active ingredients, formulation, mode of application), to environmental factors (i.e., temperature, humidity, and wind), to the hosts (i.e., the attractiveness of individual people to insects), and the insect strain (i.e., the sensitivity of the insects to repellents, biting density) [51].

In our opinion, data on the insecticide and repellent efficacy of botanicals should be critically examined when not compared with control groups consisting of synthetic analogues. For example, the myrtle essential oil showed a repellency of 62.2% against *Ph. papatasi*, which could be considered a good result, although the same sand fly strain was more susceptible to diethyl-m-toluamide (i.e., 87%) [47].

The data herein analyzed suggests that plant-based products used as insecticides should be a promising alternative in theory, but the lack of field studies makes it difficult to say whether they are a real alternative to chemical products.

On the contrary, considering the repellent properties of these compounds considerably changes the scenario; in fact, these products, although they show a repellent activity that have reached excellent results in some cases, have a rather short duration. In particular, the repellency of these products lasted only a few hours (i.e., up to a maximum of nine hours) [50] and this short duration of activity greatly limits its real use, especially if we consider the field of veterinary medicine. In fact, in the management of a serious disease such as canine leishmaniasis, nothing can be left to chance. The use of natural repellents would mean applying these products very often on the coat of the animal, which would require a high compliance of the owner.

Furthermore, to the best of our knowledge, there are no data in the literature on the safety of the repeated application of these products on dogs and/or on human skin. Indeed, there is a need for a valid alternative to insecticides and synthetic repellents; however, the road to identifying a botanical alternative is still long.

## 5. Conclusions

The current review suggests that although there is a rapidly growing body of literature on botanical insecticides and repellents against the sand fly species vectors of *Leishmania* spp., much of this literature is limited to in vitro studies conducted in laboratory conditions. Most of the studies were focused on the insecticide effect of plant-based products; environmental treatment against adult sand flies as well as the control of immature sand flies are considered unpractical stages as they have a transitory effect and are unsustainable in the long term due to several technical and economic reasons [4,7]. To date, the sole strategy recognized for the prevention of leishmaniosis is the use of topical repellent compounds; as underlined herein, only a few studies investigated the repellency of plant-based products, and these studies highlighted that natural compounds demonstrate a repellent function for a really short period of time, which could significantly limit their sustainability and use under normal conditions. The road is still long before we can say that there is a natural alternative to chemical-based repellents. Therefore, although alternatives to synthetic products are needed, a large amount of preliminary studies still need to be performed before a valid natural alternative could be formulated.

## Figures and Tables

**Figure 1 vetsci-08-00150-f001:**
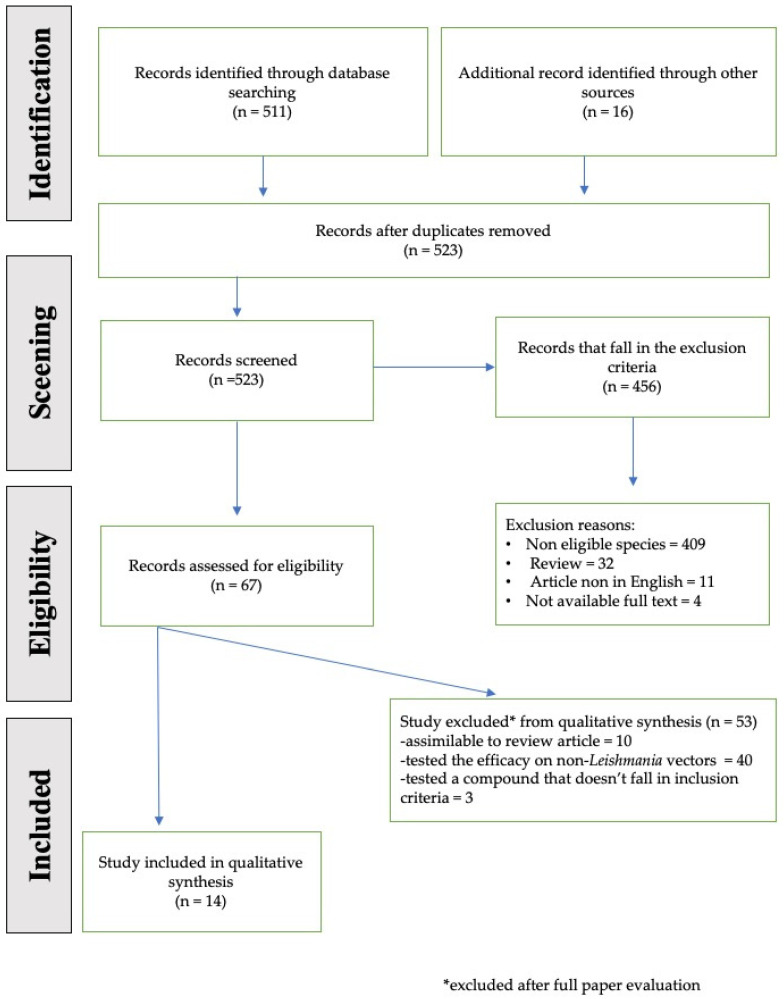
Scientific reports that fully satisfy the inclusion criteria.

**Figure 2 vetsci-08-00150-f002:**
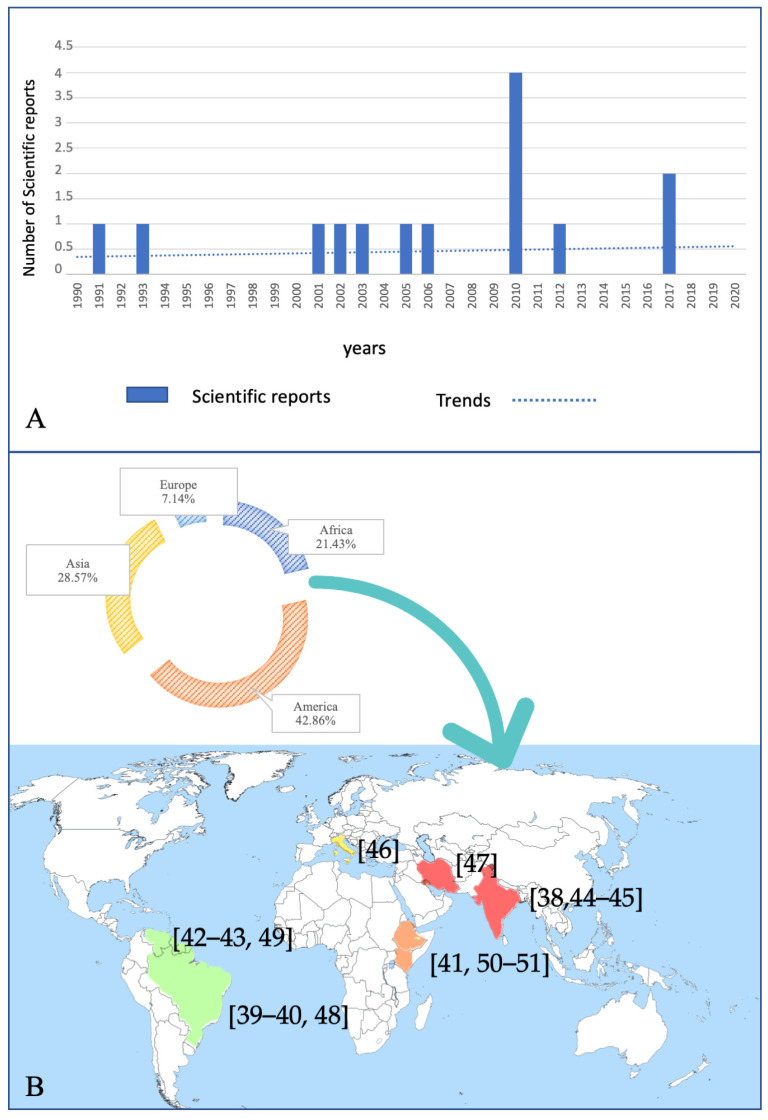
Number of scientific reports in the study period (**A**) and their geographical origin (**B**).

**Table 1 vetsci-08-00150-t001:** The table summarizes the characteristics of study examined considering the plant formulations and their efficacy.

	Scientific Name	Common Name	Formulation	mg/mL or %	Efficacy after 72 h (%)	Study Type	Sand Fly Species	*Leishmania Species*	Ref
I	*Citrus medica*	Green Lemon	Essential oil	0.01	70%	*Vivo*	*Lu. youngi*	*L. infantum*	[49]
	*Citrus medica*	Green Lemon		0.01	78%	*Vitro*	*Ph. papatasi*	*L. infantum*	
II	*Azadiracha indica*	Neem oil	Hexane-extracted oil	5%	100%	*Vivo*	*Ph. papatasi*	*L. infantum*	[44]
	*Azadiracha indica*	Neem oil		2%	96.6%				
III	*Azadiracha indica*	Neem oil	Hexane-extracted oil	2%	96.6%	*Vivo*	*Ph. papatasi*	*L. infantum*	[45]
IV	*Azadiracha indica*	Neem oil	Essential oil	0.01	75%	*Vitro*	*Ph. papatasi*	*L. infantum*	[38]
				0.02	82%				
V	*Antonia ovata*	-	Aqueous extract	223	80%	*Vitro*	*Lu. longipalpis*	*L. infantum*	[39]
	*Derris amazonica*	-		212	66.7%				
VI	*Allium sativum*	Garlic	Essential oil	0.005	40%	*Vivo*	*Ph. papatasi*	*L. infantum*	[46]
				0.01	65%				
				0.10	90%				
				1.00	95%				
VII	*Mirtus communis*	Myrtle	Essential oil	1.9	62.2%	*Vivo*	*Ph. papatasi*	*L. infantum*	[47]
VIII	*Azadirachta indica*	Neem oil	Hexane-extracted oil	2%	96.28%	*Vitro*	*Ph. orientalis*	*L. donovani*	[50]
				5%	98.26%				
	*Melia azedarach*	Persian lilac oil		2%	95.13%	*Vivo*			
				5%	96.20%				
				2%	95%	*Vitro*	*Ph. bergeroti*	*L. major*	
	*Azadirachta indica*	Neem oil		5%	95%				
				2%	92%				
				5%	92%				
IX	*Hyptis suaveolens*	Pignut	Essential oil	n.r.	No efficacy	*Vivo*	*Lu. migonei*	*L. infantum*	[48]
	*Pimenta racemosa*	West Indian bay tree							
	*Monticalia imbricatifolia*	Saccoloma							
	*Espeletia schultzii*	Frailejón							
	*Plectharanthus amboincus*	Cuban oregano							
	*Piper marginatum*	Cake bush							
	*Pseudognaphalium calciforum*	Ladies’ tobacco							
	*Cinnamomun zeylancium*	Cinnamon							
X	*Eucalyptus staigeriana*	Eucalyptus	Essential oil	0.3	1.7%	*Vitro*	*Lu. longipalpis*	*L. infantum*	[40]
				0.6	11.7%				
				1.2	32.34%				
				2.5	65.81%				
				5	100%				
	*Eucalyptus citriodora*	Lemon Scented eucalyptus		2	7.1%				
	*Eucalyptus globosus*	Southern blue gum		4	23.8%				
				6	45.2%				
				8	70%				
				10	100%				
				2	3.1%				
	*Eucalyptus globosus*	Southern blue gum		4	10.6%				
				6	25.8%				
				8	47.64%				
				10	96.47%				
XI	*Tagetes minuta*	Mexican marigold	Methanol extract	2.5	50%	*Vitro*	*Ph. duboscqi*	*L. major*	[41]
				5	63%				
				10	100%				
	*Acalypha fruticosa*	Birch-Leaved Cat Tail	Ethyl Acetate extract	2.5	60%				
				5	48%				
				10	100%				
	*Tarchonanthus camphoratus*	Camphor bush	Methanol extract	2.5	10%				
				5	10%				
				10	20%				
XII	*Monticalia greenmaniana*	Saccoloma	Essential-oil	0.001	95%	*Vitro*	*Lu. migonei*	*L. infantum*	[42]
				0.1	100%				
				0.2	-				
				0.3	-				
			Methanol extract	0.1	100%				
				1	100%				
				10	-				
				100	-				
			Aqueous extract	0.1	100%				
				1	100%				
				10	-				
				100	-				
XIII	*Argeratina jahnii*	Snakeroot	Methanol extract	0.1/1/10	No efficacy	*Vitro*	*Lu. migonei*	*L. infantum*	[43]
			Aqueous extract						
			Essential oil	0.1	22%				
				1	100%				
				10	100%				
	*Argeratina pichinchensis*	Fragrant snakeroot	Methanol extract						
			Aqueous extract						
			Essential oil						
XIV	*Cymbopogon citratus*	Lemon grass	Essential oil	0.125	51.3%	*Vitro*	*Ph. duboscqi*	*L. major*	[51]
				0.25	59.1%				
				0.50	89.1%				
				0.75	87.7%				
				1.00	100%				
	*Tagetes minuta*	Mexican marigold		0.125	21.5%				
				0.25	46.8%				
				0.50	52.2%				
				0.75	76%				
				1.00	88.9%				

**Table 2 vetsci-08-00150-t002:** Botanical products tested as insecticide and/or repellent in the studies analyzed.

	Family	Scientific Name	Number of Scientific Reports
I	*Amaryllidaceae*	*Allium sativum*	1
II	*Asteraceae*	*Monticalia imbricatifolia*	1
		*Espeletia schultzii*	1
		*Pseudognaphalium calciforum*	1
		*Cinnamomun zeylancium*	1
		*Tagetes minuta*	2
		*Tarchonanthus camphoratus*	1
		*Monticalia greenmaniana*	1
		*Argeratina jahnii*	1
		*Argeratina pichinchensis*	1
III	*Citrus*	*Citrus medica*	2
IV	*Euphorbiacee*	*Acalypha fruticosa*	1
V	*Labiatae*	*Plectharanthus amboincus*	1
VI	*Lamiaceae*	*Hyptis suaveolens*	1
VII	*Loganiacaceae*	*Antonia ovata*	1
		*Derris amazonica*	1
VIII	*Meliacee*	*Melia azedarach*	2
		*Azadirachta indica*	4
IX	*Myrtacae*	*Myrtus communis*	1
		*Pimenta racemosa*	1
		*Eucalyptus staigeriana*	1
		*Eucalyptus citriodora*	1
		*Eucalyptus globosus*	1
X	*Piperaceae*	*Piper marginatum*	1
XI	*Poaceae*	*Cymbopogon citratus*	1

## Data Availability

All data associated with this study can be obtained from the corresponding author on reasonable request.

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
