# Peer review of "Natural Products against Sand Fly Vectors of Leishmaniosis: A Systematic Review"

_vetsci, 2021, doi:10.3390/vetsci8080150_

Round 1
Reviewer 1 Report
The present manuscript carried out a systematic review study on possible phytochemical alternatives for the control of sandfly species recognized as vectors of Leishmania spp.
I have some considerations for the manuscript.
Fig. 1. Please provide the number of articles per database.
Table 1. “72hrs” – 72h
I would like to read a more developed discussion about the origin of studies. What hypothesis do the authors have to explain the largest number of studies in America and Asia? There is only one prominent country in Europe (Italy), what is the possible explanation for this?
Figure 3A confuses the reader. The x-axis with the years is incomprehensible. Prefer bar chart.
Author Response
Dear Reviewer,
We have revised the manuscript vetsci-1277268 entitled "Natural products against sand fly vectors of leishmaniosis: a systematic review" according to the Reviewers comments, who we thank for her/his valuable comments that have improved the quality and readability of the manuscript.
Here below, please find the point-by-point response to the comments.
We do hope that the revised version of the manuscript is suitable for publication in the journal.
The present manuscript carried out a systematic review study on possible phytochemical alternatives for the control of sandfly species recognized as vectors of Leishmania spp.
I have some considerations for the manuscript.
Fig. 1. Please provide the number of articles per database.
Q: We thanks the Reviewer for his/her suggestions but considering that there were no major differences between the different databases searched, we prefer to report only in the main text this information and maintain the PRISMA flow as presented in the original version of the manuscript. In the revised version of the manuscript, we specify the number of scientific reports for each database.
Q: Table 1. “72hrs” – 72h
A: Done.
Q: I would like to read a more developed discussion about the origin of studies. What hypothesis do the authors have to explain the largest number of studies in America and Asia? There is only one prominent country in Europe (Italy), what is the possible explanation for this?
A: The discussion about the origin of studies was present yet in the discussion (lines 258-276). Asia and America are published the major number of publications on plant-based products The law presence of publications in Europe, it may be due to the limited use in the tradition of products of natural origin, and primarily to the fact that in southern Europe leishmaniosis is a health problem mainly in dogs.
Q: Figure 3A confuses the reader. The x-axis with the years is incomprehensible. Prefer bar chart.
A: The figure 3A was modified according to the Reviewer suggestion.

Reviewer 2 Report
I have read with interest the manuscript “Natural products against sand fly vectors of leishmaniosis: a systematic review”
This is an interesting manuscript, however, prior to further processing of the paper several points need to be clarified.
IN MATERIAL AND METHODS SECTION
-In inclusion and exclusion criteria, please add the selected period of time
-When you say “Full text” you mean “Free full text”
- Add that the prism system has been used (www.prisma-statement.org)
IN RESULTS
In Flow Diagram, indicate records identified for each database separately
Author Response
Dear Reviewer,
We have revised the manuscript vetsci-1277268 entitled "Natural products against sand fly vectors of leishmaniosis: a systematic review" according to the Reviewers comments, who we thank for her/his valuable comments that have improved the quality and readability of the manuscript.
Here below, please find the point-by-point response to your comments.
We do hope that the revised version of the manuscript is suitable for publication in the journal.
I have read with interest the manuscript “Natural products against sand fly vectors of leishmaniosis: a systematic review”
This is an interesting manuscript, however, prior to further processing of the paper several points need to be clarified.
IN MATERIAL AND METHODS SECTION
Q: In inclusion and exclusion criteria, please add the selected period of time
R: Done
Q: When you say “Full text” you mean “Free full text”
R: No, we mean that a full text is available, regardless of it is an open access or not. Q: Add that the prism system has been used (www.prisma-statement.org)
R: Done
IN RESULTS
Q: In Flow Diagram, indicate records identified for each database separately
R: Please see the answer to Reviewer 1 comment.

Round 2
Reviewer 2 Report
In the new version of the manuscript, the authors have included suggestions indicated by the reviewer and so the article has improved substantially.